# Effect of CNTs in Copper Matrix on Mechanical Characteristics and Tribological Behavior under Dry Sliding and Boundary Lubrication Conditions

**DOI:** 10.3390/ma12132203

**Published:** 2019-07-09

**Authors:** Lin Zhao, Pingping Yao, Haibin Zhou, Taimin Gong, Minwen Deng, Zhongyi Zhang, Yelong Xiao, Hui Deng, Yang Li, Fenghua Luo

**Affiliations:** State Key Laboratory of Powder Metallurgy, Central South University, Changsha 410083, China

**Keywords:** wear mechanisms, adhesive wear, CNTs, copper, load-carrying capacity, self-lubricating composites

## Abstract

In the present work, the mechanical and tribological properties of carbon nanotube (CNT)-reinforced Cu matrix composites featuring 0–1.8 vol% CNTs prepared by spark plasma sintering under dry sliding and boundary lubrication conditions were investigated. The results of microstructure revealed that the bonding interface in Cu/CNT composites was very well established. Additionally, the addition of CNTs has a positive effect on improving the hardness and tensile strength of the composites, while the stress–strain response quasi-static tensile test confirms the same results. CNTs are proved to lead to certain self-lubrication. The addition of CNTs could result in decreased coefficient of friction (COF) and wear rates by reducing adhesive wear under dry sliding conditions, while the oil lubricating film was the major reason for decreased COF under boundary lubrication conditions.

## 1. Introduction

Copper metal matrix composites show good potential application for heavy load-carrying sliding bearing capacity [1,2,3]. Higher strength with enhanced heavy load-carrying capacity and self-lubricating ability is required for these high-performance Cu matrix composites as the service conditions become increasingly rigorous [4,5]. As we know, carbon nanotubes (CNTs) are used as ideal strengthening phases in metal matrix composites due to their unique structure and ultrahigh strength compared to metal matrix [6,7,8,9,10,11]. Many investigations concerning CNT-reinforced metal matrix composites have been performed. Some studies [12,13] have shown that due to the bridging and pulling-out effect of addition CNTs in metal matrix, mechanical properties such as hardness and tensile strength are improved significantly. In copper matrix/CNT composites prepared by different methods, consistent conclusions could also be confirmed in related studies [14,15,16]. However, the wettability and reactivity between CNTs and Cu are known to be poor [17], and the interface bonding and uniform dispersion of CNTs in all methods are not very good, especially for high CNT content [13,18]. Thus, ultrasonic dispersion followed by mechanical ball-milling [19,20] can be used to disperse CNTs in the metal matrix as a more effective technique, and spark plasma sintering (SPS) has been used since it involves rapid heating and sintering [15]. Few studies have reported on the improved interfacial characteristics of Cu/CNT composites using SPS. CNTs are one-dimensional nanofibers with a large surface area, and Cu ions have larger curvature and smaller surface area. This causes geometric incompatibility between CNTs and Cu powder, which impedes even mixing of the powders into a homogeneous composite. Thus, it is crucial to investigate the interfacial bonding in such composites to understand the reason for the improved mechanical properties.

In addition, many studies have reported and reviewed the tribological properties of metal matrix composites. Lin [21] proved that the addition of CNTs in a volume fraction ranging from 5% to 20% would cause obvious differences in the wear mechanism under dry wear tests. Rajkumar [22] reported that the coefficient of friction (COF) and wear rate increased when the volume fraction of CNTs was beyond 15%, and COF decreased under the self-lubrication effect of carbonaceous film. Tsai [23] indicated that the surface deformation of Cu/CNT composites could be reduced by adding CNTs to the Cu matrix, and a carbonaceous lubricating film was formed on the contact surface during sliding. It could be concluded from the above results that the COF and wear rate of composites under dry sliding conditions decreased by adding CNTs to the Cu matrix due to positive effect on forming a carbonaceous lubricating film.

However, there are still very few studies regarding a comparison of the tribological properties and wear mechanisms of Cu/CNTs under dry sliding and boundary lubrication conditions [16]. Moreover, few have pointed out the correlation between the tribological properties of Cu/CNT composites and their mechanical properties. Hence, Cu/CNT composites containing 0, 0.3, 0.6, 1.2, and 1.8 vol.% of CNTs were prepared in the present study. Then, the effect of additional CNTs on the mechanical properties and strengthening mechanism of the composites was analyzed. The tribological properties and wear mechanisms under dry sliding and boundary lubrication conditions are also discussed. Furthermore, the relationship between mechanical and tribological properties is illustrated by theoretical analyses and laboratory tests.

## 2. Materials and Experimental Methods 

### 2.1. Preparation of Samples

Figure 1 shows a schematic illustration of the fabrication process of Cu/CNT composites. First, commercially available multiwall CNTs (MWCNTs) (density 1.8 g/cm^3^, purity >98%; Nanolab Inc., Waltham, MA, USA) prepared by chemical vapor deposition were used as raw materials. An ultrasonic dispersion method was used to disperse CNTs evenly in alcohol for ~60 min until the CNT suspension was obtained [24,25]. Afterward, certain ratios of CNT alcohol suspension and electrolytic Cu powder (average size 38.3 μm, density 8.92 g/cm^3^) were physically mixed at a speed of 280 rpm by using a planetary ball mill for 8 h. Brass milling balls with diameters ranging from 3.8 to 6.4 mm were used, and the weight ratio of brass ball to mixture was approximately 5:1. After that, the mixture was dried. A not particularly high temperature of 80 °C and a drying time of 4 h in vacuum were selected to prevent the mixture from being oxidized. The CNTs after ultrasonic dispersion and the Cu/CNT composite powders used to prepare the specimens are shown in Figure 1. Most of the CNTs could be seen to be uniformly attached to the surface of the Cu powder or embedded closely in the Cu particle surface under magnification. Subsequently, the dried mixed powders were compacted at a pressure of 400 MPa and then processed by SPS in a graphite die at 800 °C for 45 min under pressure of 40 MPa in an argon atmosphere. Finally, the specimens were prepared after cooling in water at a cooling rate of approximately 100 °C/min. 

### 2.2. Mechanical and Tribological Properties Test

The Archimedes method, following Standard GB/T 1423-1996, was used to measure the density of the composites. The Instron 3369 machine was used to measure the tensile strength of the specimens with a strain rate of 2.1 × 10^−3^ s^−1^ (crosshead speed, 1 mm/min). The tensile specimens were machined by wire electrode cutting with a cross-section of 2 × 2 mm^2^ and a gauge length of 8 mm. Vickers hardness test was selected to evaluate the hardness of the Cu/CNT composites. A frictional test was carried out using a ball-on-block tribometer (Figure 2) under dry sliding and boundary lubrication conditions, where predrip oil was used to ensure boundary lubrication. A steel ball bearing (AISI52100; 0.95–1.05 C, 0.20–0.40 Mn, 0.15–0.35 Si, 1.30–1.65 Cr; S ≤ 0.020, P ≤ 0.027, Mo ≤ 0.10, Ni ≤ 0.30, Cu ≤ 0.25, Ni + Cu ≤ 0.50) hardened to 63–65 HRC with a 9.5 mm diameter was selected as a testing ball. The samples were ground and polished using SiC grit paper until a surface roughness (Ra) of 1.2 μm was achieved. The specimens were subsequently cleaned using anhydrous alcohol. A constant normal load with a range of 5 to 60 N, a fixed sliding distance of 5 mm, a reciprocating speed of 600 rpm (0.1 m/s), and a test time of 900 s were programmed during testing. Five indentations under same conditions were performed for each composite.

The microstructure of the specimens and the worn surface of the wear tracks were characterized by scanning electron microscopy (SEM; Nova NanoSEM230, FEI, Hillsboro, OR, USA) and transmission electron microscopy (TEM; JEM-2100F, JEOL, Tokyo, Japan) at a working voltage of 200 kV. The average length (mm) and cross-sectional area (mm^2^) of wear tracks were measured by a 3D surface profilometer (Nano Map 500-LS, AEP Technology, Braselton, GA, USA), then multiplied to get the wear volume. Then the wear rate was calculated utilizing Equation (1)
*W* = *V*/(*f* · *L*)(1)
where *W* is the wear rate, *V* is the wear volume (mm^3^), *f* is the friction force (N), and *L* is the sliding distance(m) of specimen. 

## 3. Results and Discussion

### 3.1. Microstructure of Cu/CNT Composites

It could be seen from the metallographs that pure Cu is more compact than Cu/0.6 CNT composite (composition: 99.4 vol% Cu and 0.6 vol% CNT, referred to as Cu/0.6 CNT) and twin crystals were seen in both, as shown in Figure 3a,b. Note that the dark regions in Figure 3b, corresponding to pores, indicate that porosity increases after the addition of CNTs. Due to the scale difference, the nanoscale CNTs could not be clearly observed. However, the uniform homogeneous distribution of the metallographic structure of Cu/CNT composites suggests that CNTs are probably uniformly distributed in composites [26,27]. In order to clearly describe the effect of additional CNTs on the density of Cu/CNT composites, the relative densities of pure Cu and Cu/CNT composites are shown in Figure 3c. The density decreased as the volume fraction of CNTs increased. The relative density of the Cu/CNT composite was lower than that of pure Cu, as expected. 

Figure 4 shows TEM images of the Cu/CNT composite. As shown in Figure 4a, the SEM image demonstrates the existence of a transitional interfacial layer at the interface bonding region and the damaged CNTs after ball-milling. In order to determine the microstructure of the interfacial layer, high-resolution TEM (HRTEM) images and corresponding energy-dispersive X-ray spectroscopy (EDS) analysis were carried out. As shown in Figure 4b, the grain boundary can be clearly observed. The interfacial layer zone displays the lattice fringes of the Cu-(200) and Cu-(111) faces with spacing of 0.18 nm and 0.20 nm, as shown in the fast Fourier-transform (FFT) image (Figure 4c). This suggests that some Cu atoms appear among the CNTs, presumably causing deformation of the CNTs and forming a coherent lattice with them [26,27]. The EDS analysis (marked B in Figure 4a) shown in Figure 4d also confirms that the element composition of Cu and C did not undergo a component mutation, but there was a gradient change in the region of the CNT and Cu interface, which is in agreement with HRTEM results. Therefore, it can be speculated that CNTs and Cu matrix have a good bonding interface. Generally, the ball-milling process is able to destroy the structures of the outer surfaces of CNTs via repetitive deformation, cold welding, and fracturing of the mixed powders [24,28]. However, carbon bonds were broken in the prism planes of the outer surfaces of damaged CNTs as shown in Figure 4a. As a result, this promotes the diffusion of Cu to form good interfacial bonding. Hence, a strong interface bonding that can transfer loads effectively is a significant factor in improving the mechanical properties of the composites [29].

### 3.2. Mechanical Properties 

Figure 5a shows the typical quasi-static stress–strain curves of pure Cu and Cu/0.6 CNT composite. All the specimens mentioned in this paper presented the similar mechanical responses to with that seen in Figure 5a when exposed to quasi-static loads. It can be seen from the curve shape that the specimens went had gone through the elastic deformation stage and yield stages in the tensile test. However, a much higher modulus and larger failure strain are observed in Cu/0.6 CNT composite. CNTs with high strength and modulus take loads and inhibit the deformation of the material during the process of tensile strength testing. Therefore, Cu/CNT composites present higher modulus, tensile strength, elongation, and failure strain than pure Cu. The tensile strength and Vickers hardness of the specimens are shown in Figure 5b. First, it can be observed that all composites show higher tensile strength and Vickers hardness than pure Cu; both of them increased first and then decreased as the content of CNTs increased, such similar phenomena were observed in aluminum /CNTs composites [30]. Therefore, the addition of CNTs can enhance the tensile strength and Vickers hardness of copper matrix. The Cu/0.6 CNT composite exhibited higher tensile strength (308 MPa) and Vickers hardness (106 Hv) than pure Cu. A lower relative density but a higher tensile strength and Vickers hardness for Cu/CNT composites can be ascribed to the high stiffness and strength of CNTs. When the external loads act on the Cu /CNT composites, it is easy to transfer through the bonding interface, consequently improving the hardness and tensile strength of the composite material with a uniform distribution of CNTs. The significantly increased strength also indicates that the composites were successfully fabricated by this combination process. However, a further increase in CNT content led to increased porosity, as well as the possibility of CNTs agglomerating in the specimens. The rising porosity lowered the hardness, whereas the agglomeration of CNTs reduced the contact area and weakened the interface bonding between CNTs and matrix, inevitably resulting in a decline in tensile strength.

The typical fractured surfaces of as-prepared composites after tensile tests are shown in Figure 6. The toughening nest, which tends to gradually increase in size as CNTs increase, is found in the fracture morphology of all Cu/CNT composites, implying that ductile fracture occurs during the test. In the enlarged fracture morphology, the CNT clusters are observed locally in the toughening nest and pores. Moreover, many single CNTs are attached to the surface of the Cu matrix, and some Cu particles are embedded in CNTs clusters, as shown in Figure 6d. It can be predicted that the amount of pull-out CNTs increases as the volume fraction increases, such phenomenon was also found in previous studies [14,15,28]. In addition, a CNT bridge can be observed. When the matrix reached its ultimate heavy load capacity during the test, CNTs also played a load-bearing role by suppressing the matrix crack extension, thus promoting crack deflection [29]. This is beneficial to improve the heavy load-carrying capacity of the material with higher strength. When the volume fraction is greater than or equal to 1.2%, CNT clusters appear in the fracture, indicating that the CNTs are poorly dispersed in the matrix, leading to weak bonding in the composite. It is worth pointing out that CNT clusters and degraded bonding were the main reasons for the decline in tensile strength [9].

Generally, Orowan strengthening and the load-transfer effect are proposed to explain the strengthening phenomenon observed in particle-reinforced metal matrix composites [29]. Combining TEM observations with mechanical properties, the improvement in Cu/CNT composites could be due to two points: (a) homogeneous dispersion of CNTs in the Cu matrix. CNTs dispersed in composites can effectively hinder grain growth during the heating process, thus improving the properties of the composite. (b) Strong interfacial bonding strength can effectively transfer loads when the external loads work.

### 3.3. Tribological Properties under Dry Sliding and Boundary Conditions

Figure 7 shows the average COF and corresponding wear rate of the specimens with a load of 10 N under dry sliding conditions. It can be seen that the Cu/0.6 CNT composite exhibited the lowest COF and wear rate, which shows that it has the best antiwear ability. The COF and wear rate declined when the fraction of added CNTs was lower than 0.6 vol% but slowly increased when the proportion went above 0.6 vol%. Note that the change in COF and wear rate is contrary to that in hardness and tensile strength (see Figure 5b) as the volume fraction of CNTs increases, indicating that the tribological behavior is closely related to the mechanical properties of composites. The increased hardness will lead to a decreased direct metal contact area. Thus, the Cu/0.6 CNT composites with the highest hardness show the lowest COF. Moreover, according to the test standard variation, pure Cu presents higher fluctuations in COF than other composites in this work. This can be ascribed to severe adhesion at the two contact surfaces and work-hardening of pure Cu. However, as more CNT particles are incorporated into the Cu matrix, the average COF of Cu/CNT composite becomes remarkably lower and more stable during tests. The positive effect of the improved mechanical properties by adding CNTs could lead to lower wear track depth during tests [31]. As the CNT content increases, the direct contact area on the counterpart will be reduced during the friction test. Consequently, the probability of adhesive wear will decrease. What is more, the addition of CNTs may enable the composite to have a certain self-lubricating capacity [32]. All of these factors confirm that the addition of CNTs can improve antiwear ability and reduce the tendency of adhesive wear [12,32,33]. Furthermore, the addition of CNTs may give the material a certain self-lubricating capacity. However, more CNTs added into the Cu matrix leads to more CNT clusters in the composite, which reduces the heavy load capacity and wear resistance.

Figure 8 shows the worn surfaces. The adhesion spalling pits (Figure 8a) were characterized as typical features in pure Cu, due to strain-hardening caused by the reciprocating friction force, and cracks induced worse plastic deformation ability. At the same time, friction led to the formation of fresh surfaces and the occurrence of adhesive wear, which was observed in other Cu matrix composites [2,3]. With increased CNTs, the worn surfaces became quite different (see Figure 8b–e). Adhesion spalling pits gradually disappeared while cracks, grooves, and fatigue pits were found on the worn surface. Compared to other Cu/CNT composites, the worn surface of Cu/0.6 CNT composites in Figure 8c presents typical features of grooves, quite different from those of Cu/0.3 CNT composites shown in Figure 8b. The grooves were believed to be generated by hard asperities on the counterpart and reciprocal wear debris plowing and grinding over the composite. Because of the improved mechanical properties, enhanced adhesive resistance, and plastic deformation resistance while adding CNTs, the wear mechanism changed gradually from adhesive to plow wear. It could be observed that the worn surfaces in Figure 8d,e are rougher and the grooves are deeper and wider than those in Figure 8c. The fatigue pits in Figure 8e indicate the main wear mechanism by plow wear and fatigue wear, and the decreased mechanical properties caused by the high CNT content may be the key reason for this phenomenon. Therefore, it can be confirmed again that the addition of CNTs should not exceed 0.6 vol%. Figure 8f show the higher magnification worn surface of Cu/CNTs composite, and the EDS results of the marked region in Figure 8f indicate that it contains a higher content of carbon than other regions, and the morphology of this strip is similar to that of added CNTs. Since the composite consists only of Cu and CNTs, it can be confirmed that thin strips of CNTs can obviously be seen to be distributed parallel to the sliding direction. Therefore, it is difficult for cracks and large debris drops to form, resulting in less wear [34].

Figure 9 shows the average COF curves of Cu/CNT composites with a pressure range from 5 N to 60 N under boundary conditions. Obviously, the COF of the composites first decreased and then increased as the volume fraction of CNTs increased. The Cu/1.2 CNT composite exhibited the lowest COF. Compared to the average COF under dry test conditions, the variation trend of the friction curve was basically the same, and the average COF decreased significantly as expected, but the CNT content in the composite with the lowest COF changed from 0.6 to 1.2 vol%. The reduced COF with this range of CNTs could be attributed to the lubricating oil film at the sliding surface, which can effectively prevent direct contact between two sliding surfaces. There may also be two other factors: first, a positive effect on the mechanical properties with a lower CNT content might have an important role in the descent stage of COF. Second, adding excessive CNTs deteriorates mechanical properties, such as the heavy load capacity and wear resistance, thus raising the COF in the ascent stage. Additionally, the pores to store lubricating oil might be the major reason for the lowest COF change from Cu/0.6 CNT to Cu/1.2 CNT composites.

According to the adhesive theory [35], friction force (*f*) is composed of shear force on the adhesive part of the shear surface and shear force on the lubricating film, and can be expressed as Equation (2). The load (*F_n_*) is supported by contact of the microconvex body and lubricating film, and can be expressed as Equation (3):(2)$$f=A_{r}\left[\alpha \tau_{b}+\left(1-\alpha \right)\tau_{f}\right]$$
(3)$$F_{n}=A_{r}\left[\alpha \delta_{s}+\left(1-\alpha \right)\delta_{sf}\right]$$ where f is the friction force, $A_{r}$ is the total actual area under load, α is the percentage of direct metal contact area over total area under load, $\tau_{b}$ is the shear strength of soft metals, $\tau_{f}$ is the shear strength of lubricant film, $F_{n}\;$is the load, $\delta_{s}$ is the compressive yield strength limit of the material, and $\delta_{sf}$ is the yield limit of lubricant film. Then the COF (*µ*) can be written as
(4)$$µ=\frac{f}{F_{n}}=\frac{\alpha \tau_{b}+\left(1-\alpha \right)\tau_{f}}{\alpha \delta_{s}+\left(1-\alpha \right)\delta_{sf}}$$

Under oil lubricating conditions, α is very small, and $\delta_{s}$ and $\delta_{sf}$ are almost equal, so$\;µ=\tau_{f}/\delta_{s}$. Generally, the shear strength of lubricating film is much smaller than that of metal, so the COF under oil lubricating conditions is much smaller than that of dry friction. In addition, it is worth noting that the role of improved mechanical properties is non-negligible in the decrease of α, which reduces the COF. Otherwise, the COF will rise. This explains the different COF under oil lubricating conditions with different volume fraction CNTs from a theoretical perspective.

The worn surface topography of Cu/CNT specimens under oil lubricating conditions is shown in Figure 10. It can be seen from Figure 10a–e that the worn surfaces of the Cu composites with different volume fractions of CNTs are totally different. However, the change trend is similar to the dry friction condition except for the obvious adhesive wear feature. Composites with CNTs in the range of 0.3 to 1.2 vol% (see Figure 10b–d) have better heavy load capacity than other Cu/CNT composites under the current working conditions. Too much or too little CNT content is detrimental to the tribological properties, leading to severe plastic deformation, as indicated by a large number of cracks or fatigue pits, seen in Figure 10a,e. Thin strips of CNTs are seen to be distributed parallel to the sliding direction in Figure 11f.

To better understand how CNTs affect the COF, wear rate, and wear mechanism of composites, we used a schematic illustration of wear mechanism of Cu/CNT composites under dry friction and boundary lubrication conditions, shown in Figure 11a,b, respectively. Under the same loading pressure (F_n_ in Figure 11) and dry friction conditions, as shown in Figure 11a, the depth of the counterpart pressed into the composite (h in Figure 11) and the contact area will decrease due to the improved mechanical properties by adding CNTs, thus reducing the friction force (f), which prevents motion. Therefore, the COF and wear rate of Cu/CNT composite were smaller than those of pure Cu. Under boundary lubrication conditions, as shown in Figure 11b, the oil lubricating film carried most of the load and its low shear strength might be the major reasons for the lower COF and wear rate than those of the dry friction condition, and the mechanical properties had similar effects on h and the contact area under dry friction conditions. It would not be repeated again here.

In terms of wear mechanism, it is well known that there are many microconvex bodies on the contact surface, and, under dry friction conditions, the pressure load causes them to have direct contact, as shown in the magnified image in Figure 11a. Then the reciprocating F_f_ causes severe plastic deformation of the microconvex bodies. This then leads to the initiation, development, and fracture of cracks, and thus produces wear debris. It is easy to form adhesive pits because of the obvious adhesive tendency of the counterpart and pure Cu, while in Cu/CNT composites, the CNTs on the worn surface were arranged along the sliding direction with the plastic deformation, as shown in Figure 11a. This may be the main reason for the decreased adhesive wear tendency, resulting in lower COF and better wear resistance. Compared to the adhesive mechanism in pure Cu, plowing is the main wear mechanism in Cu/CNT composites, caused by the reciprocating motion of the microconvex bodies and the third body on the worn surface. Under boundary lubrication conditions, the existence of the lubricating oil film greatly reduces the microconvex body direct contact area, as shown in Figure 11b. This avoids the possibility of adhesive wear and greatly reduces the plastic deformation of the composite under Ff compared with that under dry friction. Thus, the phenomenon of parallel arrangement of CNTs along the sliding direction was less than in dry friction conditions, as shown in Figure 11b. Therefore, both pure Cu and Cu/CNT composites are dominated by the plow wear mechanism. It is also worth pointing out that the worn surface topography results in this study are consistent with the above wear mechanism, and the volume fraction of CNTs is closely related to the significance of the wear mechanism.

## 4. Conclusions

Cu/CNT composites were prepared by SPS combined with ultrasonication treatment and mechanical ball-milling before consolidation. The effects of CNTs on mechanical and tribological properties were investigated. The following conclusions can be drawn.

(1) Uniformly dispersed carbon nanotubes in the Cu matrix can be obtained through ultrasonic dispersion and mechanical ball-milling. The relative density of the Cu/CNT composite was close to that of pure Cu.

(2) The addition of CNTs can effectively improve the mechanical properties of the composites. Cu/0.6 CNTs exhibited the best hardness and tensile strength. The CNTs and Cu matrix were well bonded at the interface, with Cu atoms appearing in the CNTs locally.

(3) The lowest COF and wear rates of Cu/CNT composites were obtained when the CNTs equaled 0.6 vol.% under dry sliding conditions and 1.2 vol.% under boundary lubrication conditions. The CNTs also had certain self-lubricating properties.

(4) The wear mechanisms under dry friction and boundary lubrication conditions were different. The CNTs mainly reduced COF and wear rates by reducing the adhesive wear tendency under dry friction conditions, while the oil lubricating film was the major reason for the decreased COF under boundary conditions. Composites with CNT volume fraction between 0.6% and 1.2% exhibited the best tribological performance and heavy load carrying capacity.

## Figures and Tables

**Figure 1 materials-12-02203-f001:**
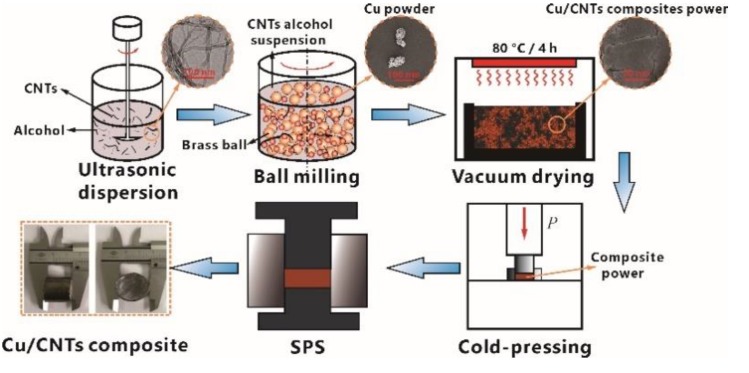
Schematic illustration of the fabrication process of Cu/carbon nanotube (CNT) composite.

**Figure 2 materials-12-02203-f002:**
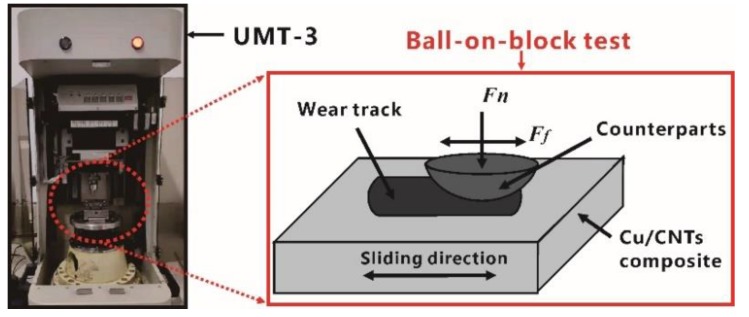
Schematic illustration of ball-on-block tribometer.

**Figure 3 materials-12-02203-f003:**
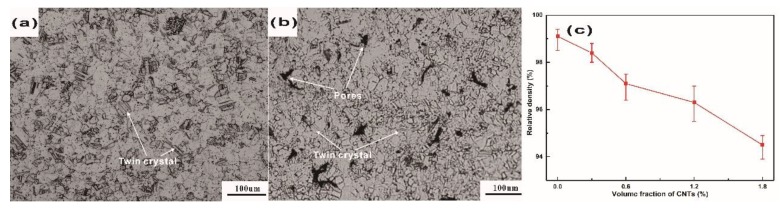
Typical metallographs of (**a**) pure Cu and (**b**) Cu/0.6 CNTs and (**c**) relative density of Cu/CNT composite.

**Figure 4 materials-12-02203-f004:**
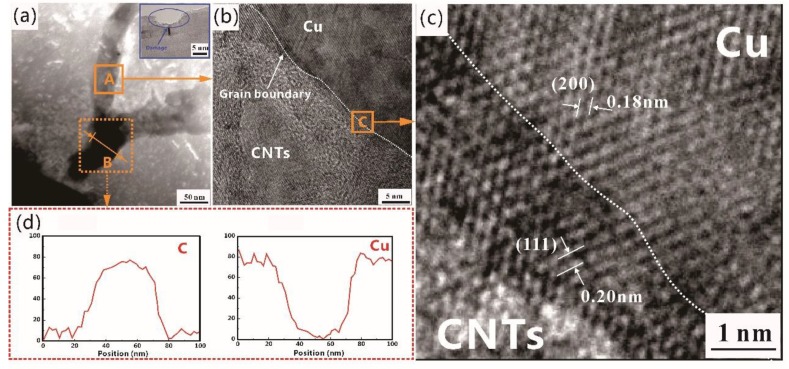
TEM images of Cu/CNT composite: (**a**) interface bonding of Cu and CNTs, (**b**) high-resolution TEM (HRTEM) image of highlighted region A, (**c**) HRTEM image of highlighted region C, and (**d**) energy-dispersive x-ray spectroscopy (EDS) line-scanning results of region B.

**Figure 5 materials-12-02203-f005:**
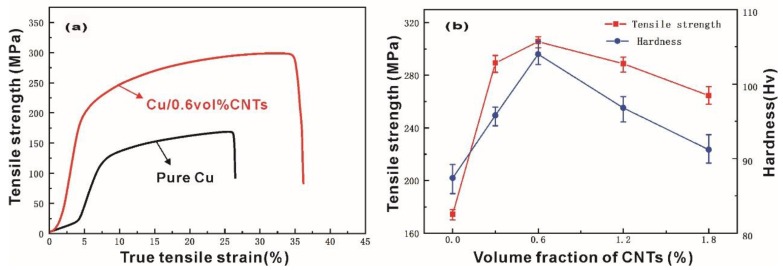
Mechanical properties of pure Cu and Cu/CNT composites: (**a**) typical quasi-static stress–strain curves and (**b**) tensile strength and Vickers hardness.

**Figure 6 materials-12-02203-f006:**
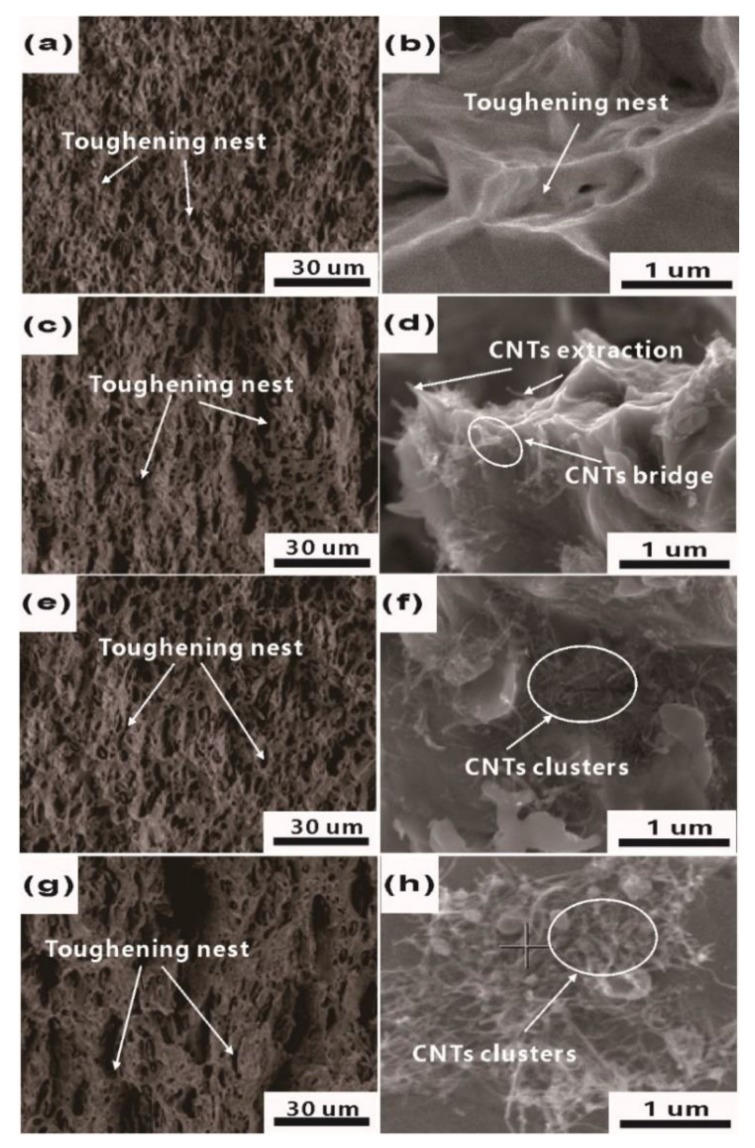
SEM micrographs of fractured surfaces of Cu/CNT composites containing (**a**,**b**) 0.3, (**c**,**d**) 0.6, (**e**,**f**) 1.2, and (**g**,**h**) 1.8 vol.% CNTs after tensile tests.

**Figure 7 materials-12-02203-f007:**
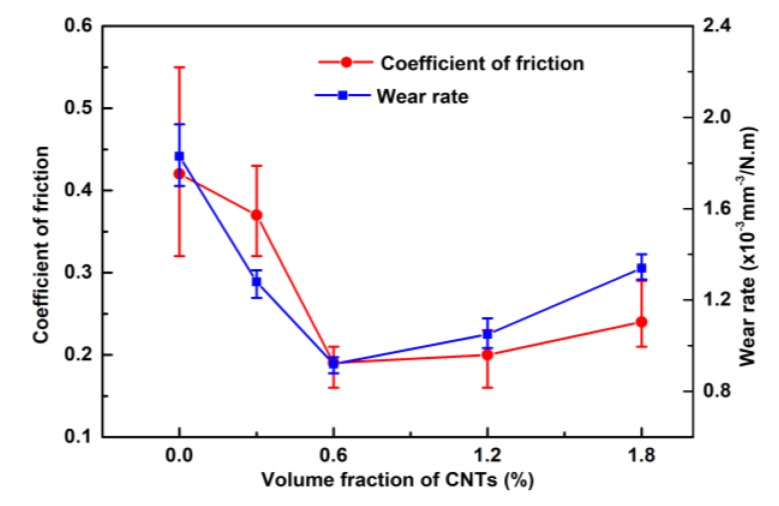
Coefficient of friction (COF) and wear rates of pure Cu and Cu/CNT composites with a pressure of 10 N under dry sliding conditions.

**Figure 8 materials-12-02203-f008:**
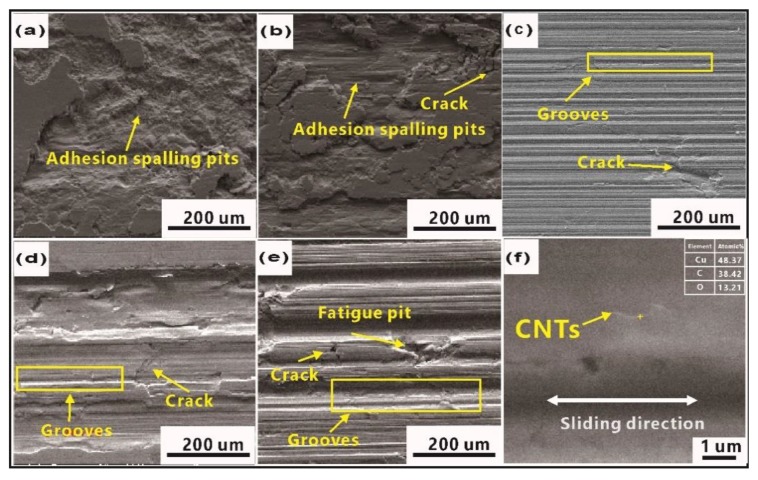
SEM images of worn surface under dry friction conditions of Cu/CNT composites containing: (**a**) 0, (**b**) 0.3, (**c**) 0.6, (**d**) 1.2, and (**e**) 1.8 vol.% CNTs and (**f**) highly magnified image of worn surface.

**Figure 9 materials-12-02203-f009:**
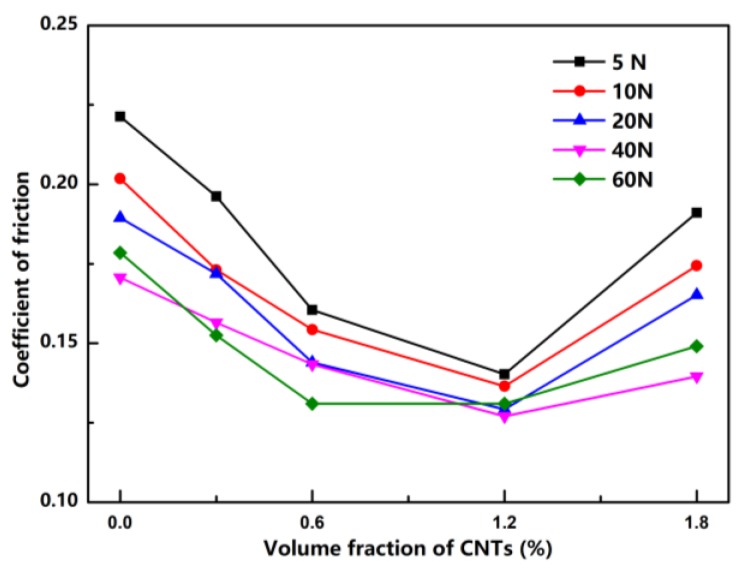
Average COF of pure Cu and Cu/CNT composites under different loading pressures and boundary lubrication conditions.

**Figure 10 materials-12-02203-f010:**
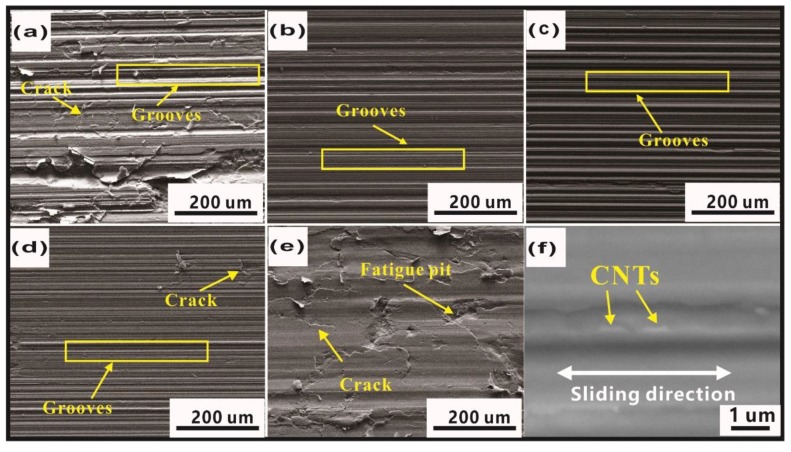
SEM images of worn surface under boundary lubrication conditions of Cu/CNT composites containing: (**a**) 0, (**b**) 0.3, (**c**) 0.6, (**d**) 1.2, and (**e**) 1.8 vol.% CNTs under a load of 40 N and (**f**) highly magnified image of typical grooves.

**Figure 11 materials-12-02203-f011:**
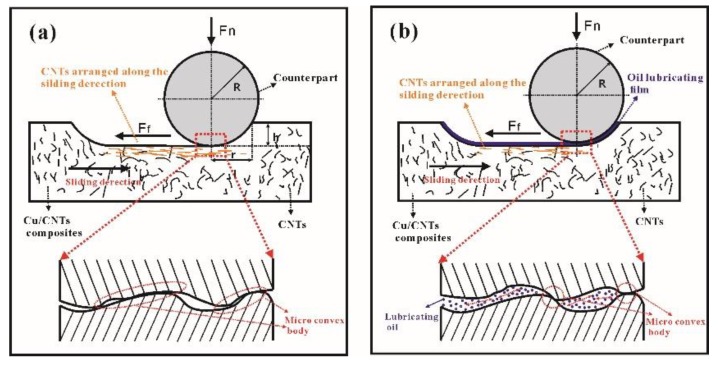
Schematic illustration of wear mechanism: (**a**) dry friction condition and (**b**) boundary lubrication condition.

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
