# Peer review of "Effect of CNTs in Copper Matrix on Mechanical Characteristics and Tribological Behavior under Dry Sliding and Boundary Lubrication Conditions"

_materials, 2019, doi:10.3390/ma12132203_

Reviewer 1 Report

materials-531511: Effect of CNTs in copper matrix on mechanical characteristic and tribological behavior under dry sliding and boundary lubrication conditions, Lin Zhao, Pingping Yao, Haibin Zhou, Taimin Gong, Minwen Deng, Zhongyi Zhang, Yelong Xiao, Hui Deng, Yang Li, Fenghua Luo.

Technical comments (not in order of importance):

Comment #1: I am concerned about quality and integrity of the CNTs used in the research. The authors conclude that the improvement in the mechanical and tribological properties is due to ultra-high strength and the unique structure of the CNTs. Contrary to this claim, the tensile strength of many types of CNTs is reported to be low compared with we expect (Materials Research Express, 6, Article number 055047, 2019.) How do reader understand this contradictory result?

Comment #2: It is reported that friction coefficient of the composite containing 0.6CNTs has reached 0.2. Interestingly, this value is similar with the value obtained from SWCNT solid. Furthermore, Raman intensity ratio is good index for the evaluation of carbon materials abundance. Takagi et al, (Journal of Nanoscience and Nanotechnology, 8, 2665-2670, 2008) used Raman technique to investigate structural change of CNTs before and after friction tests, and concluded that formation of CNT-derived transferred films lead to induce lower friction coefficient characteristics. I expect that Takagi’s work yields valuable information for strengthening this manuscript.

Comment #3: Regarding the relationship between CNT content and mechanical behavior, the relationship reported in this paper is very similar to the results of Hashida et al (Materials Science and Engineering A, 617, 179-186, 2014.) That is, do these result have a trade-off relationship between improvement and degradation in the mechanical properties with respect to the volume fraction of the CNTs

External comments:

Comment #4: Please describe specimen preparation procedures, geometrical information and testing condition of the tensile tests.

Comment #5: Please describe how the authors measured wear volume (V). Would you like to include a cross-sectional surface profile of the friction surface in this paper?

Comment #6: Please add density values of Cu and CNTs.

Comment #7: I recommend the paper for publication after revision, which takes into account the specific comments given in the reviews.

Reviewer 2 Report

This paper is poorly written.  It requires completely rewritten and major revisions

1. Line 46: Introduction is confusing, in between authors included about Al matrix composites. What is the connection with Al and Cu matrix.

Results [11-12] showed that the addition of CNTs to Al matrix could improve the mechanical properties of the composites. In addition, many results [13-15] confirmed that the mechanical properties such as tensile strength, hardness and elongation of Cu/CNTs composites were 50 improved through different prepared method

2. Line 67: Rajkumar [21] reported that the COF and wear rate increased when the volume fraction of CNTs beyond 15%, and coefficient of friction (COF) decreased under the self-lubrication effect of carbonaceous film. Abbreviation should be defined first, not later.

3. Fig. 4 caption is confusing due to a,b,c…, please modify the caption for clear understanding.

4. Line 161: Authors are stating that the carbon bonds were broken in the prism planes of the outer surfaces of damaged CNTs. How do authors confirm that the carbon bonds are broken in the prism plane?

5. line 173: Authors are stating that “both composites present similar mechanical response”. Which are both composites? Do the authors mean pure Cu and Cu/0.6CNT, if yes then please mention on what phenomena/property authors are stating that the mechanical response is similar.

6. Line 190: as CNTs content further increased, lower hardness in the composite might arise in the weaker regions where the sliding of grains could take place during test.

Please mention what is weaker region? And what is sliding of grains?

7. line 201: The amount of pull-out CNTs increases as the volume fraction of CNTs increases. In addition, the CNTs bridge can also be observed.

How do the authors quantify/see the pull out CNTs?

8. Line 236: The positive effect of the improvement of the mechanical properties by adding CNTs could lead a lower pressure depth during tests. Please explain what is lower pressure depth.

9. Fig.8(f) How can author confirm the presence of CNT?

10. Arrange the reference number. In Page 13, line 292 Reference #1 is not acceptable.

11. Define all the symbols used in equation 3-1, 3-2, 3-3

 Author Response

Round  2

Reviewer 2 Report

Authors have incorporated several suggestions. However, for most of the questions, instead of answering the questions, authors have simply removed the sentences from the manuscript.  This is highly unacceptable.  Authors must provide the rebuttal for all the questions raised by the reviewer in the previous version.

In addition to this authors need to comment on the following

Fig.8(f) How can the author confirm the presence of CNT? 

The authors have not provided sufficient proof for the questions.

Also, define all the symbols used in equation 3-1, 3-2, 3-3

Symbols used in the figures are confusing. For examples figure 4.
